# Reduced Kidney Function and Relative Hypocalciuria—Observational, Cross-Sectional, Population-Based Data

**DOI:** 10.3390/jcm9124133

**Published:** 2020-12-21

**Authors:** Massimo Cirillo, Giancarlo Bilancio, Pierpaolo Cavallo, Francesco Giordano, Gennaro Iesce, Simona Costanzo, Amalia De Curtis, Augusto Di Castelnuovo, Licia Iacoviello

**Affiliations:** 1Department of Public Health, University of Naples Federico II, 80131 Naples (NA), Italy; 2Department Scuola Medica Salernitana, University of Salerno, 84081 Baronissi (SA), Italy; giancarlo.bilancio@gmail.com (G.B.); francesco.giordano1989@gmail.com (F.G.); iesceg@gmail.com (G.I.); 3Department of Physics, University of Salerno, 84084 Fisciano (SA), Italy; pcavallo@unisa.it; 4Istituto Sistemi Complessi, Centro Nazionale Ricerche, 00185 Rome (RM), Italy; 5Department of Epidemiology and Prevention, IRCCS Neuromed, 86077 Pozzilli (IS), Italy; simona.costanzo@neuromed.it (S.C.); amalia.decurtis@moli-sani.org (A.D.C.); licia.iacoviello@uninsubria.it (L.I.); 6Mediterranea Cardiocentro, 80122 Napoli (NA), Italy; dicastel@ngi.it; 7Department of Medicine and Surgery, Research Center in Epidemiology and Preventive Medicine (EPIMED), University of Insubria, 21100 Varese (VA), Italy

**Keywords:** kidney function, eGFR, calcium, parathyroid hormone, 1,25-dihydroxyvitamin D, epidemiology

## Abstract

This observational, cross-sectional, epidemiological analysis investigated relationships of kidney function to urine calcium and other variables. The analyses targeted two population-based samples of adults (Gubbio study and Moli-sani study: *n* = 3508 and 955, respectively). Kidney function was assessed as estimated glomerular filtration rate (eGFR). Calcium/creatinine ratio (Ca/Cr) was used as index of urinary calcium in timed overnight urine under fed condition (Gubbio study), morning urine after overnight fast (Gubbio study), and first-void morning urine (Moli-sani study). Moli-sani study included also data for glomerular filtered calcium load, tubular calcium handling, and serum phosphorus, parathyroid hormone, 1,25-dihydroxyvitamin D, calcium, and 25-hydroxyvitamin D. eGFR positively and independently related to Ca/Cr (*p* < 0.001). In multivariate analyses, eGFR lower by 10 mL/min × 1.73 m^2^ related to overnight urine Ca/Cr lower by 14.0 mg/g in men and 17.8 mg/g in women, to morning urine Ca/Cr lower by 9.3 mg/g in men and 11.2 mg/g in women, and to first-void urine Ca/Cr lower by 7.7 mg/g in men and 9.6 mg/g in women (*p* < 0.001). eGFR independently related to glomerular filtered calcium load (*p* < 0.001) and did not relate to tubular calcium handling (*p* ≥ 0.35). In reduced eGFR only (<90 mL/min × 1.73 m^2^), low urine Ca/Cr independently related to low serum 1,25-dihydroxyvitamin D (*p* = 0.002) and did not relate to hyperphosphatemia, high serum parathyroid hormone, or hypocalcemia (*p* ≥ 0.14). Population-based data indicated consistent associations of lower kidney function with lower urine calcium due to reduction in glomerular filtered calcium. In reduced kidney function, relative hypocalciuria associated with higher prevalence of low serum 1,25-dihydroxyvitamin D.

## 1. Introduction

The term ‘chronic kidney disease—mineral and bone disorder’ (CKD–MBD) is used to define a complex syndrome secondary to the chronic decline in kidney function that encompasses the progressive appearance of biochemical, skeletal, and cardiovascular abnormalities [1]. The CKD-MBD biochemical abnormalities include the reduction of 1,25-dihydroxyvitamin D [1,25(OH)_2_D], the increase in serum phosphorus and serum parathyroid hormone (PTH), and the possible reduction in serum calcium (Ca) [1]. The current guidelines recommend the monitoring of the serum lab tests for early diagnosis and control of CKD-MBD [1]. The guidelines do not mention urine tests although, at least theoretically, low serum concentrations of 1,25(OH) _2_D together with high serum concentrations of PTH could contribute to secondarily reduced urine Ca [2]. Two studies reported data regarding the relationship between kidney function and urine Ca [3,4]. In 2007, Craver et al. reported a positive, non-linear relationship of kidney function with 24 h urinary Ca in a subgroup of 319 nephropathic patients with glomerular filtration ranging from <15 to ≥90 mL/min × 1.73 m^2^ as assessed by creatinine clearance predicted using the Cockcroft-Gault formula [3]. In 2019, Ramalho et al. reported a positive, linear relationship of kidney function with 24 h urinary Ca in 365 patients with estimated glomerular filtration rate (eGFR) ranging from ≥15 to 59 mL/min × 1.73 m^2^ [4]. Two other studies retrospectively investigated urinary Ca and kidney function in kidney stone(s) patients [5,6], hence in patients in whom the high prevalence of hypercalciuria could play a major confounding [7]. Data on the association between kidney function and urinary Ca are missing in the general population. Data are missing also for urine collected under fasting conditions to assess the possible role of the intake or absorption of Ca [8]. The present study analyzed two sets of epidemiological data to investigate the relationship of kidney function with urinary Ca in general population. It also reports the consistence of results under fed and fasting conditions, and in different population samples, the relationship of kidney function with classical indices of kidney Ca handling, and the possible use of urine Ca as proxy of biochemical abnormalities typical of CKD-MBD.

## 2. Methods

The Gubbio study is a population-based cohort study ongoing since 1982 in the city of Gubbio, Italy [9]. The study adheres to the Declaration of Helsinki of 1975, as revised in 2013, and included an informed consent and the approval by the institutional committee (CEAS-Umbria #2850/16). Study design, main exams, response rates, and characteristics of the Gubbio cohort were previously reported [9]. The present analysis deals with data collected at the second exam of the study (1989–92) [9]. The exam included the following: timed collection of overnight urine under fed conditions from the first void after the completion of the evening dinner to the first void at morning wake-up included [10]; early morning blood sample collected after an overnight fast and after the completion of the overnight urine collection [9,10,11]; morning urine collection under fasting conditions performed after blood sampling [11]; medical visit for measurements of anthropometry and blood pressure and for administration of questionnaires [9]. Urine concentrations of Ca and creatinine were measured in fresh samples using automated biochemistry and quality controls [9]. Serum creatinine was measured in frozen samples by automated biochemistry (Express Plus, Bayer Diagnostic) using a kinetic alkaline picrate assay with IDMS-traceable standardization [12]. Variability in blind duplicates was <5% for all these measurements [9]. Target cohort for the present analysis consisted of 4670 examinees who, at the second exam of the Gubbio study, were aged ≥18 years and with complete data for eGFR and overnight urine. Of these examinees, 1162 were excluded because of missing morning urine collection under fasting conditions. Thus, the Gubbio study dataset included 3508 adults with complete data both for overnight urine and for morning urine under fasting conditions. Excluded examinees had similar overnight urine Ca/creatinine ratio and slightly lower values for female sex prevalence, age, body mass index, and eGFR (Appendix A).

The Moli-sani study is a prospective cohort study ongoing since 2005 that enrolled 24,325 individuals from 2005 to 2010, men and women, aged ≥35 years, randomly recruited from the general population of a region of central-southern Italy [13]. The study complies with the Declaration of Helsinki of 1975, as revised in 2013, and was approved by the Rome Catholic University ethical committee (P99, A.931/03-138-04, 11 February 2004). All participants provided written informed consent. The baseline visit included questionnaires about socioeconomic status, physical activity, medical history, dietary habits, risk factors, personal and family medical history; measurements of blood pressure and anthropometry; collection of untimed urine spot samples from the first void at wake up; collection of morning venous blood samples after an overnight fast. Biological samples were processed for lab tests within 3 h and/or stored in liquid nitrogen as described [14]. Lab tests for the whole cohort included the measurements of serum cystatin C [15]. Target cohort for the present analysis was a sub-group of 1000 examinees of the Moli-sani study that were selected by a sex- and age- stratified randomization for additional lab tests using frozen samples of serum and urine [16]. The stratification was designed to have 100 men and 100 women for each one of the following five age-groups: 35–44, 45–54, 55–64, 65–74, and ≥75 years. Additional lab tests included the measurements of the urine concentrations of creatinine, Ca, and phosphorus, and the measurements of serum concentrations of creatinine, total Ca, albumin, phosphorus, total 25-hydroxy-vitamin D [25(OH)D], PTH, and 1,25(OH) _2_D [16]. Automated biochemistry was used for the measurements of serum creatinine, Ca, albumin, phosphorus, PTH and for the measurements of urine creatinine, and Ca (Abbott, IL, USA) [16,17]. Serum creatinine was measured by enzymatic assay calibrated with IDMS-traceable standard [12]. The measurements of 25(OH)D and 1,25(OH) _2_D were performed by a fully automated chemiluminescent assay (Diasorin, Saluggia, Italy) [16,18]. As previously described and in accordance with guidelines [16,19], the 25(OH)D assay was calibrated using ID-LC-MS- and ID-LC-MS/MS- traceable standard NIST-SRM 972a [20]. Intra- and inter- assay variability of chemiluminescent methods in blind duplicates was <5% [16]. The date of visit that was used to calculate the average local solar global horizontal irradiance in the month prior to blood withdrawal (from here on defined as solar irradiance) [21]. This variable was used to control the analyses for the possible effect of ultraviolet exposure on serum 25(OH)D [22]. Of the subgroup selected for additional lab test, 45 examinees were excluded because of missing data. Thus, the Moli-sani study dataset included 955 adults with complete data for urine and serum.

### 2.1. Measurements in the Gubbio Study Dataset

Kidney function was assessed as eGFR calculated by the Chronic Kidney Disease—Epidemiology Collaboration equation with the use of serum creatinine, ethnicity, sex, and age [23]. Both in overnight and in morning fasting urine, urine Ca was assessed as urine Ca/creatinine ratio [24] to exclude the errors in timing and completeness inevitable in timed collections [25]. Previous papers showed that overnight urine Ca excretion rate highly correlated with 24-h urinary Ca [26,27]. Appendix A shows the correlation between overnight Ca/creatinine ratio and the estimated ratio for 24 h urine. The list of variables in analyses included also anthropometry, estimated 24 h urinary creatinine [28], ending time of evening meal, duration of overnight fast from completion of the evening meal to initiation of morning fasting urine collection, reported habitual intake of milk or yogurt [29] as index of the major dietary source of absorbable Ca [30], and reported treatment with Ca or vitamin D supplementation.

### 2.2. Measurements in the Moli-Sani Study Dataset

Kidney function was assessed as eGFR calculated by the Chronic Kidney Disease—Epidemiology Collaboration equation including serum creatinine, ethnicity, sex, age, and cystatin C also to reduce the confounding of creatinine generation [31,32]. Urine Ca was assessed as urine Ca/creatinine ratio similarly to the Gubbio dataset. For the investigation of kidney Ca handling, the following indices were calculated: serum albumin-bound Ca = serum albumin as g/100 mL times the multiplier 0.88 [33]; serum ultra-filterable Ca = serum total Ca minus serum albumin-bound Ca; glomerular filtered Ca load = serum ultra-filterable Ca times eGFR; fractional Ca excretion = urine Ca/serum ultra-filterable Ca times serum creatinine/urine creatinine [34]; fractional tubular Ca reabsorption = 1 − fractional Ca excretion [34]. For investigation on metabolic abnormalities of CKD-MBD [1], hypocalcemia was defined as serum total Ca < 8.6 mg/100 mL, hyperphosphatemia as serum *p* ≥ 4.5 mg/100 mL, high serum PTH as serum PTH ≥ 66 pg/mL, and low serum 1,25(OH) _2_D as 1,25(OH) _2_D < 18 pg/mL [35].

### 2.3. Statistical Analyses

eGFR was the main independent variable in both datasets and was divided in the following seven strata: ≥90, 89–75, 74–60, 59–45, 44–30, 29–15, and <15 mL/min × 1.73 m^2^. eGFR < 90 mL/min × 1.73 m^2^ was defined as reduced kidney function [36]. First, the relationship of eGFR to urinary Ca was investigated by univariate ANOVA of Ca/creatinine ratio along eGFR strata for overnight urine (Gubbio study dataset), morning fasting urine (Gubbio study dataset), and first-void morning urine (Moli-sani study dataset). The independence of results from urine creatinine was investigated by multi-variable ANOVA and linear regression with control for age and weight that, together with sex, are the main predictors of creatinine generation [28]. As ancillary data, the Appendix A includes additional analyses in the Gubbio study dataset for overnight urine calcium excretion rate, estimated 24 h urinary Ca calculated as overnight urine Ca/creatinine times 24 h urinary creatinine, and Ca/creatinine ratio in overnight urine of the 1162 examinees excluded from analyses due to missing morning urine collection under fasting conditions. After that, in the Moli-sani dataset, the relationships of eGFR to indices of kidney Ca handling and to metabolic indices of CKD-MBD were investigated using univariate and multi-variable ANOVA along eGFR strata. Finally, analyses in the Moli-sani dataset investigated the association of urine Ca/creatinine ratio with the prevalence of CKD-MBD metabolic abnormalities separately in the group without reduced kidney function and in the group with reduced kidney function. To this aim, chi-square analysis, ANOVA, and logistic regression were performed along the following four strata of urine Ca/creatinine ratio: ≥100, 99–75, 74–50, and <50 mg/g. All analyses were done separately in men and women. Statistical procedures were performed using IBM-SPSS 19. Results were reported as prevalence for categorical variables, mean ± SD for numerical non-skewed variables, median with interquartile range (IQR) for numerical skewed variables, regression coefficient, odds ratio (OR), and 95% confidence interval (95%CI).

## 3. Results

### 3.1. Descriptive Statistics

Table 1 shows descriptive statistics by sex in the two datasets. Mean age was approximately 10-year higher in the dataset of the Moli-sani study that did not enroll individuals with age 18–34 years but only individuals with age ≥35 years. In both datasets, men and women differed for anthropometry, urine and serum creatinine, eGFR, urine Ca, urine creatinine, and Ca/creatinine ratio. In the Gubbio study dataset, Ca/creatinine ratio was higher in overnight urine than morning urine after overnight fast. In both datasets, no examinee had eGFR < 15 mL/min × 1.73 m^2^. In both datasets, there were positive trends along eGFR strata for female sex prevalence, age, and body mass index (Appendix A).

### 3.2. Kidney Function and Urine Ca

In the Gubbio study dataset, lower eGFR related to lower Ca/creatinine ratio in overnight urine and morning fasting urine (Figure 1, upper and central panels). The relationship was similar in first-void urine of the Moli-sani study dataset (Figure 1, lower panels). In both datasets, the relationships were more linear with adjustment for age and weight (Figure 1).

Results in the Gubbio study dataset were similar also for overnight urinary Ca excretion rate and estimated 24 h urinary Ca (Appendix A), and for overnight urine Ca/creatinine ratio in the subgroup of 1162 examinees excluded from main analyses (Appendix A).

In linear regression with control for age and weight, a difference in eGFR of 10 mL/min × 1.73 m^2^ associated with a difference in overnight urine Ca/creatinine ratio of 14.0 mg/g in men and of 17.8 mg/g in women (95%CI = 11/17 and 15/21, *p* < 0.001), with a difference in morning fasting urine Ca/creatinine ratio of 9.3 mg/g in men and of 11.2 mg/g in women (95%CI = 7/12 and 9/13, *p* < 0.001), and with a difference in first-void urine Ca/creatinine ratio of 7.7 mg/g in men and of 9.6 mg/g in women (95%CI = 4/12 and 5/15, *p* < 0.001). Results were similar in analyses controlled also for additional variables (Appendix A).

### 3.3. Kidney Function and Kidney Ca Handling

In the Moli-sani study dataset, lower eGFR linearly related to lower glomerular filtered load of Ca also after adjustment for age (Figure 2, upper panels). eGFR did not significantly relate to fractional tubular reabsorption and to fractional Ca excretion (Figure 2).

### 3.4. Kidney Function and Metabolic Markers of CKD-MBD

Figure 3 and Figure 4 summarize the results of analyses about the relationships of eGFR with serum metabolic markers of CKD-MBD in the Moli-sani study dataset. Lower eGFR associated with borderline significant trend to higher serum phosphorus in women only and with significant trends to higher PTH and to lower 1,25(OH)_2_D in both sexes (Figure 3).

For serum Ca, lower eGFR associated with weak trends to lower total Ca and higher ultra-filterable Ca (Figure 4). The contrasting trends between total Ca and ultrafilterable Ca were due to the association of lower eGFR with lower serum albumin (Appendix A). Lower eGFR did not associate with lower 25(OH)D (Figure 4). Findings for 25(OH)D were not significant also after control for solar irradiance (*p* = 0.577) that was the strongest correlate of 25(OH)D (standardized regression coefficient, beta = 0.253, *p* < 0.001).

### 3.5. Urine Ca and CKD-MBD Metabolic Abnormalities

In the Moli-sani study data set, men and women had different prevalence of hyperphosphatemia (1.2% and 7.1%, *p* < 0.001) but similar prevalence of high serum PTH (1.2 and 1.9%, *p* = 0.386), of low serum 1,25(OH)_2_ (3.1% and 2.6%, *p* = 0.638), and of hypocalcemia (1.0% and 0.6%, *p* = 0.517). In uni-variate analyses, reduced kidney function associated not significantly with higher prevalence of high serum PTH, low serum 1,25(OH)_2_D and hypocalcemia among men, and with higher prevalence of hyperphosphatemia, high serum PTH, and low serum 1,25(OH)_2_D among women (Table 2).

The relation of urine Ca/creatinine ratio to prevalence of CKD-MBD metabolic abnormalities was absent in the group without reduced kidney function whereas it was significantly inverse in men and women with reduced kidney function (Figure 5).

When univariate analyses in the group with reduced kidney function were done for single CKD-MBD metabolic abnormality, lower urine Ca/creatinine ratio related to higher prevalence of low 1,25(OH)_2_D in univariate analysis (OR for 25 mg/g lower Ca/creatinine ratio in men and women: = 2.35 and 1.93, 95%CI = 1.22/4.49 and 1.27/2.93, *p* < 0.01). Findings were not significant for the relation of urine Ca/creatinine ratio to hyperphosphatemia, high serum PTH, and hypocalcemia (Appendix A).

## 4. Discussion

This observational, population-based, cross-sectional analysis reported two main findings. First, there is a continuous, graded, and independent association of lower kidney function with lower urine Ca and, as expected, with lower glomerular filtered load of Ca, in the absence of alterations in the tubular Ca handling. Second, in individuals with reduced kidney function, the relative hypocalciuria independently associates with increased prevalence of low serum 1,25(OH)_2_D.

The interpretation of observational, cross-sectional data should be cautious. Study results indicated that a reduction in the glomerular filtered Ca load was the sole mechanism responsible for the relative hypocalciuria associated with reduced kidney function, at least over the eGFR range explorable in the two population samples in analysis. This interpretation was coherently supported by different observations: the consistence of results in urine samples under fed condition and after prolonged fast, the independence of findings of dietary markers, Ca supplementation, and duration of overnight fast. The lack of altered tubular Ca handling in reduced kidney function appeared coherent with the concurrence of conditions capable either to up-regulate and to down-regulate tubular Ca reabsorption as expected for higher serum PTH and of lower serum 1,25(OH)_2_D in combination with higher fibroblast growth factor 23 [2,37]. In a finalistic view, the relative hypocalciuria associated with lower kidney function could be interpreted as the alteration through which Ca deficiency was prevented or limited in the presence of reduced serum 1,25(OH)_2_D secondary to kidney hypofunction.

Regarding metabolic indices of CKD-MBD, study results were coherent with the concept that a chronic reduction in kidney function induces an increase in serum levels of phosphorus and PTH and a decrease in serum 1,25(OH)_2_D [1,2]. For serum Ca, study results indicated associations of lower kidney function with slightly lower total Ca and slightly higher ultra-filterable Ca due to the association of lower kidney function with slightly lower values of estimated albumin-bound Ca. Last, study results did not indicate alterations of serum 25(OH)D in reduced kidney function and confirmed the effects of solar irradiance on serum 25(OH)D [22].

Study limitations were the use of the Ca/creatinine ratio as index of urinary Ca, the lack of 24-h urine collection, the lack of direct measurements of serum ultra-filterable Ca, the lack of standardized Ca intake, the low number of individuals with severe eGFR reduction, and the lack of data for different ethnic groups. The bias due to the use of the Ca/creatinine ratio was reasonably minor considering that the results in the Gubbio cohort for overnight urine were substantially identical with the use of the urine Ca/creatinine ratio and of the timed urine Ca excretion rate. The use of overnight urine instead of 24-h urine could have caused a bias due a circadian variation in creatinine excretion although overnight urine Ca strongly correlated with same day 24-h urine Ca also with urine Ca factored by urine creatinine [27].

The study had some merits. The population-based design reduced the bias due to low sample size. The use of datasets from temporally and geographically separated epidemiological projects reduced the possibility of local or temporal confounders and proved the reproducibility of the findings. The collection of urine under different conditions proved that findings were independent of the time and the type of urine sample. The use of Ca/creatinine ratio might have ruled out the bias due to inaccuracies in timing and completeness of urine collections [25]. Lastly, in the Moli-sani data set, the study was based on the most accurate non-invasive index of glomerular filtration [31] and on the reference calibration for serum 25(OH)D measurements [19,20].

Regarding the association of reduced kidney function with relative hypocalciuria, study results were in accordance with those derived from clinical settings where data were collected under different experimental conditions [3,4]. For biochemical markers of CKD-MBD, study results were in accordance with previous evidence reporting consistent associations of reduced kidney function with higher serum phosphorus, higher serum PTH, and lower serum 1,25(OH)_2_D and less consistent association with serum Ca [1,36,38]. Finally, study results were in contrast with the report of an association of kidney function with serum 25(OH)D [39] and were in accordance with the previously reported lack of association [36,40,41].

In conclusion, this observational, cross-sectional, epidemiological study showed a continuous, graded, and independent relationship between kidney function and urine Ca. The relationship was reproducible in two different population samples, independently of the timing and of the type of the urine collection. The relative hypocalciuria associated with reduced kidney function appeared due to a reduction in the normal glomerular filtered Ca load. It could represent a beneficial adaptation to prevent or to reduce Ca deficiency secondary to the 1,25(OH)_2_D deficiency associated with reduced kidney function. Data indicated that the assessment of urine Ca in patients with reduced kidney function could be of use as a supportive evidence of low 1,25(OH)_2_D and/or for monitoring treatment with calcium or vitamin D.

The main practical implication of the present study is that the assessment of urine Ca could be added to the list of non-invasive analytes used in the diagnosis and monitoring of CKD-MBD [1]. In particular, the assessment of urine Ca could suggest the presence of low 1,25(OH)_2_D given that, in individuals with reduced kidney function, the prevalence of this metabolic abnormality was progressively higher with reducing the level of urine Ca independently of sex, age, weight, and eGFR. In addition, low urine Ca in reduced kidney function highlights the need of being cautious in the use of Ca supplementation to avoid excessively positive Ca balance in patients with reduced kidney function [42].

## Figures and Tables

**Figure 1 jcm-09-04133-f001:**
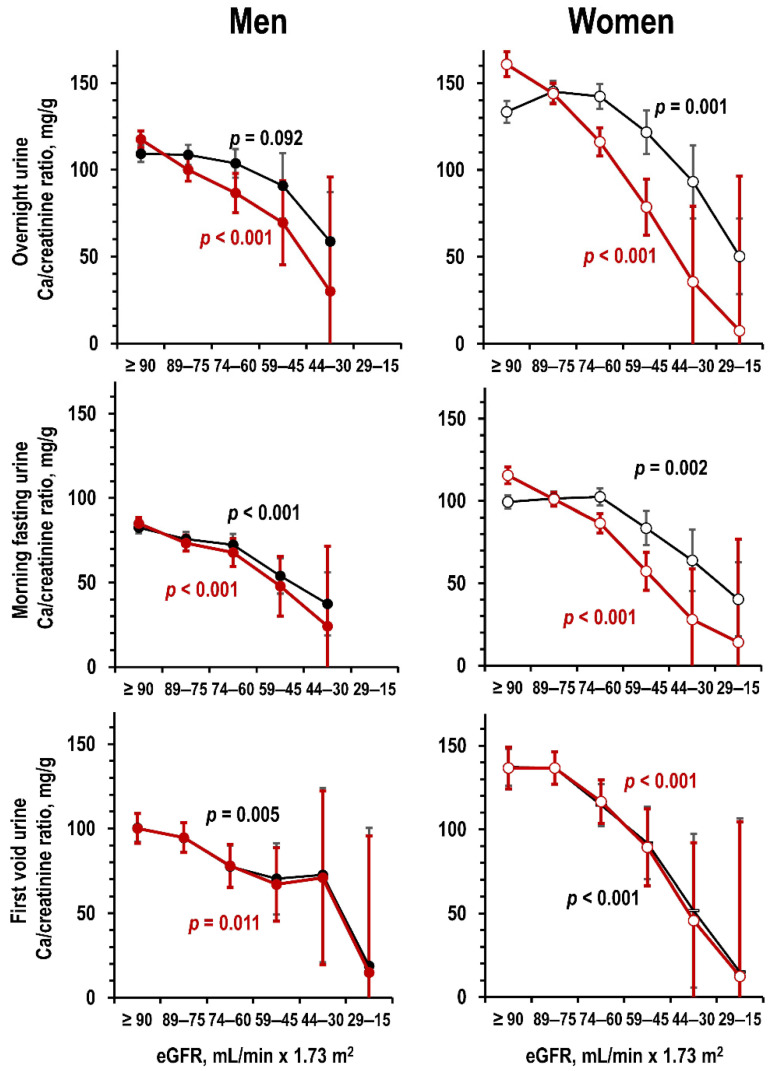
Mean and 95%CI of Ca/creatinine ratio in men and women (**left** panels and **right** panels, respectively) by eGFR stratum in overnight urine (**upper** panels, Gubbio study dataset), in morning fasting urine (**central** panels, Gubbio study dataset), and in first-void morning urine (**lower** panels, Moli-sani study dataset) in uni-variate ANOVA and in ANOVA controlled for age and weight (black line and red line, respectively). Number of individuals per eGFR stratum from left to right: Gubbio study dataset, men = 905, 451, 164, 31, 4, and 0—women = 680, 677, 471, 109, 13, and 3; Moli-sani study dataset, men = 195, 168, 88, 30, 5, and 2—women = 139, 175, 107, 36, 8, and 2. No individual had eGFR < 15 mL/min × 1.73 m^2^.

**Figure 2 jcm-09-04133-f002:**
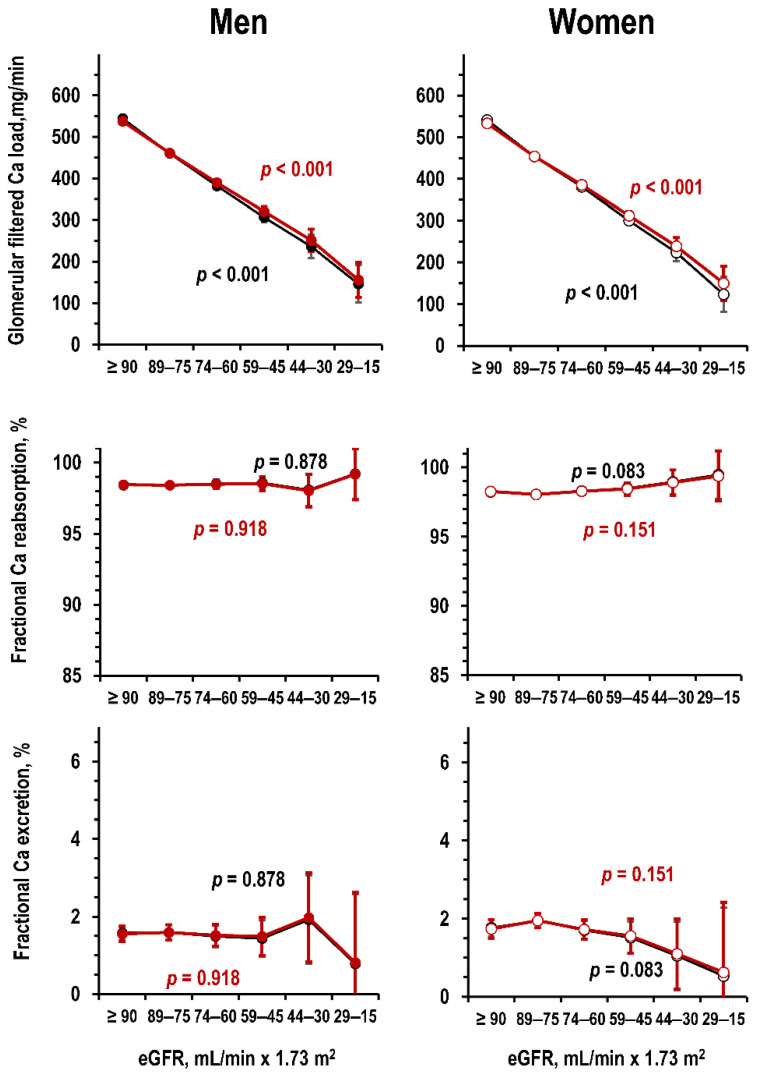
Moli-sani dataset: mean and 95%CI in men and women (**left** panels and **right** panels, respectively) by eGFR stratum of glomerular filtered load of Ca (**upper** panels), of fractional tubular reabsorption of Ca (**central** panels), and of fractional excretion of Ca (**lower** panels) in uni-variate ANOVA and in ANOVA controlled for age (black line and red line, respectively). Number of individuals per eGFR stratum from left to right: men = 195, 168, 88, 30, 5, and 2—women = 139, 175, 107, 36, 8, and 2. No individual had eGFR < 15 mL/min × 1.73 m^2^.

**Figure 3 jcm-09-04133-f003:**
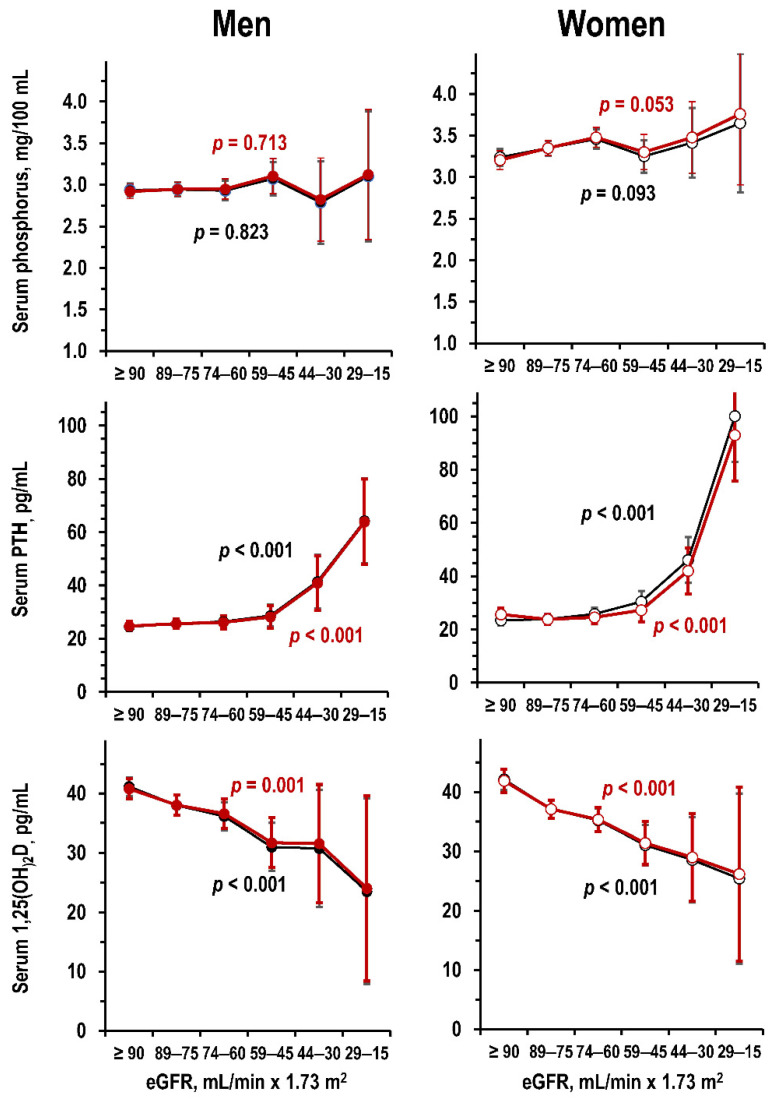
Moli-sani dataset: mean and 95%CI in men and women (**left** panels and **right** panels, respectively) by eGFR stratum of serum phosphorus (**upper** panels), serum PTH (**central** panels), and serum 1,25(OH)_2_D (lower panels) in uni-variate ANOVA and in ANOVA controlled for age (black line and red line, respectively). Number of individuals per eGFR stratum from left to right: men = 195, 168, 88, 30, 5, and 2—women = 139, 175, 107, 36, 8, and 2. No individual had eGFR < 15 mL/min × 1.73 m^2^.

**Figure 4 jcm-09-04133-f004:**
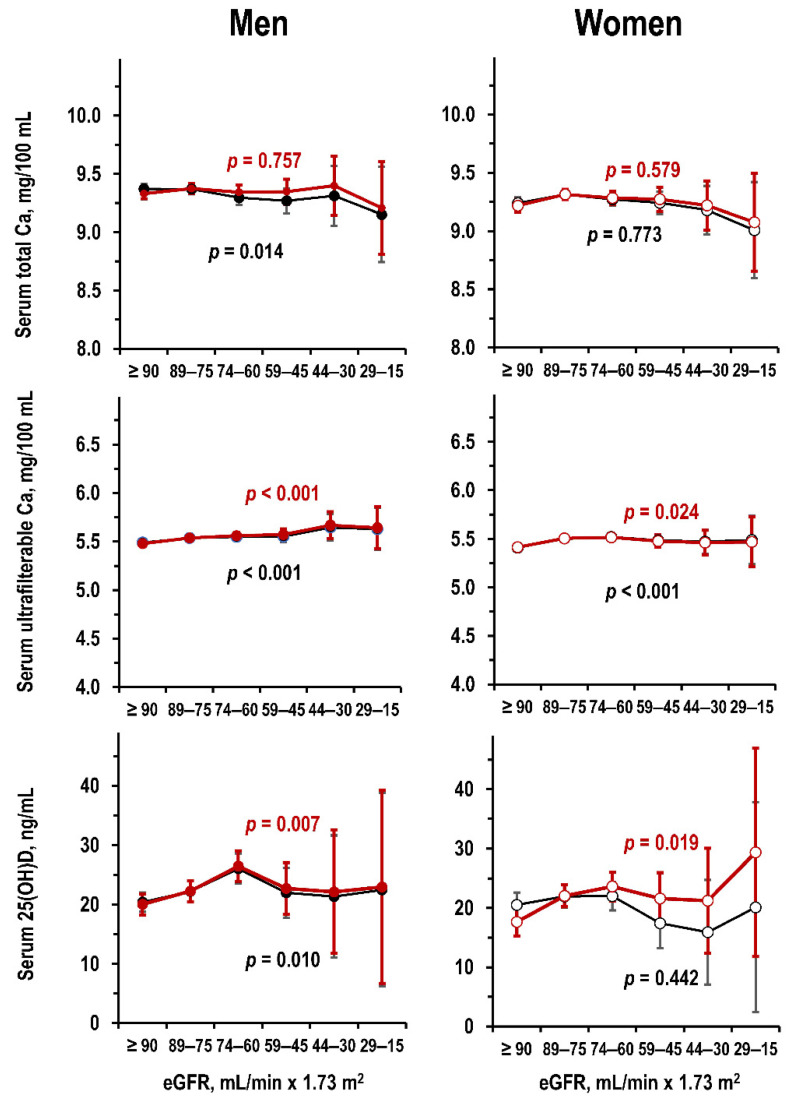
Moli-sani dataset: mean and 95%CI in men and women (**left** panels and **right** panels, respectively) by eGFR stratum of serum total Ca (**upper** panels), serum ultrafilterable Ca (**central** panels), and serum 25(OH)D (**lower** panels) in uni-variate ANOVA and in ANOVA controlled for age (black line and red line, respectively). Number of individuals per eGFR stratum from left to right: men = 195, 168, 88, 30, 5, and 2—women = 139, 175, 107, 36, 8, and 2. No individual had eGFR < 15 mL/min × 1.73 m^2^.

**Figure 5 jcm-09-04133-f005:**
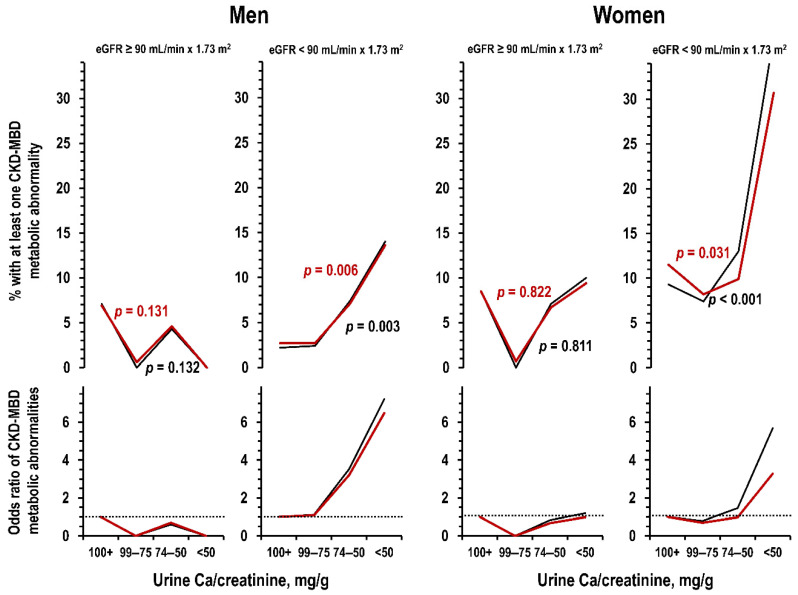
Moli-sani dataset: prevalence (**upper** panels) and OR (**lower** panels) of individuals with at least one CKD-MBD metabolic abnormality in men and women (**left** panels and **right** panels, respectively) by stratum of urine Ca/creatinine ratio, in the group without reduced kidney function and in the group with reduced kidney function. Reduced kidney function was defined as eGFR < 90 mL/min × 1.73 m^2^. *p* value are for trend along strata of urine Ca/creatinine ratio in univariate analyses and in analyses controlled for age and weight (black line and red line, respectively). Number of individuals per urine Ca/creatinine stratum from left to right: men = 176, 72, 114, and 126—women = 276, 75, 60, and 56.

**Table 1 jcm-09-04133-t001:** Descriptive statistics in the Gubbio study dataset and the Moli-sani study dataset: prevalence, mean ± SD for numerical non-skewed variable, and median (IQR) for numerical skewed variables ^.

	Gubbio Study		Moli-Sani Study	
	Men	Women	Men	Women
Number of examinees	1555	1953	488	467
Demographics and anthropometry				
Age, years	49.4 ± 16.8	51.0 ± 16.7	59.7 ± 9.6	59.9 ± 10.1
Weight, kg	77.0 ± 11.4	64.8 ± 11.3	80.8 ± 13.2	69.2 ± 13.3
Body mass index, kg/m^2^	26.9 ± 3.7	26.6 ± 4.8	28.6 ± 4.2	28.7 ± 5.5
Kidney function				
Serum creatinine, mg/100 mL	0.96 (0.89/1.03)	0.83 (0.77/0.89)	0.88 (0.80/0.97)	0.72 (0.66/0.80)
Serum cystatin C, mg/L	n.a.	n.a.	0.99 (0.88/1.13)	0.98 (0.87/1.10)
eGFR, mL/min × 1.73 m^2^	92.2 ± 15.7	83.4 ± 16.5	84.6 ± 15.6	81.5 ± 16.4
% eGFR 89–60 mL/min × 1.73 m^2^	39.5%	58.8%	52.4%	60.4%
% eGFR < 60 mL/min × 1.73 m^2^	2.3%	6.5%	7.5%	9.8%
Urinary variables				
Estimated urinary creatinine, g/24 h	1.54 ± 0.18	1.00 ± 0.16	1.52 ± 0.18	1.00 ± 0.17
Overnight urine Ca, mg/L	129 (92/182)	121 (82/172)	n.a.	n.a.
Overnight urine creatinine, g/L	1.44 (1.01/1.98)	100 (67/150)	n.a.	n.a.
Overnight urine Ca/creatinine ratio, mg/g	108 ± 68	139 ± 82	n.a.	n.a.
Morning fasting urine Ca, mg/L ^	110 (76/147)	102 (69/136)	n.a.	n.a.
Morning fasting urine creatinine, g/L ^	1.61 (1.14/2.14)	118 (77/166)	n.a.	n.a.
Morning fasting urine Ca/creatinine ratio, mg/g ^	79 ± 48	100 ± 57	n.a.	n.a.
First-void morning urine Ca, mg/L ^	n.a.	n.a.	42 (21/81)	31 (15/59)
First-void morning urine creatinine, g/L ^	n.a.	n.a.	0.68 (0.32/1.13)	0.31 (0.17/0.65)
First-void morning urine Ca/creatinine ratio, mg/g ^	n.a.	n.a.	92 ± 60	126 ± 68
Kidney Ca handling				
Glomerular filtered Ca load, mg/min	n.a.	n.a.	467 ± 85	446 ± 87
Tubular fractional Ca reabsorption,%	n.a.	n.a.	98.8 (98.1/99.2)	98.5 (97.8/99.0)
Fractional Ca excretion,%	n.a.	n.a.	1.21 (0.80/1.90)	1.51 (0.98/2.20)
CKD-MBD metabolic markers				
Serum phosphorus, mg/100 mL	n.a.	n.a.	2.94 ± 0.56	3.34 ± 0.61
Serum PTH, pg/mL	n.a.	n.a.	23.5 (17.9/30.4)	23.0 (17.1/29.5)
Serum 1,25(OH)_2_D, pg/mL	n.a.	n.a.	38.4 ± 11.6	37.5 ± 10.9
Serum 25(OH)D, ng/mL	n.a.	n.a.	22.2 ± 11.9	21.1 ± 12.8
Serum total Ca, mg/100 mL	n.a.	n.a.	9.35 ± 0.30	9.27 ± 0.30
Serum albumin, g/L	n.a.	n.a.	43.5 ± 2.6	43.1 ± 2.5
Serum ultra-filterable Ca, mg/100 mL	n.a.	n.a.	5.52 ± 0.16	5.48 ± 0.18
Other variables				
% with Ca supplementation	0.8%	0.9%	0.6%	1.7%
% with vitamin D supplementation	0.6%	1.2%	0.4%	4.9%
Habitual intake of milk or yogurt, mL/day	12 (0/125)	63 (0/125)	n.a.	n.a.
Ending time of evening meal, hour:min	20:30 (20:00/21:00)	20:50 (20:00/21:00)	n.a.	n.a.
Duration of the overnight urine collection, h	8.2 ± 1.1	8.3 ± 1.0	n.a.	n.a.
Fast duration before morning fasting urine, h	10.1 ± 1.2	10.2 ± 1.1	n.a.	n.a.
Duration of morning fasting urine collection, min	63 ± 26	68 ± 23	n.a.	n.a.
Solar irradiance, MJ/m^2^ per day	n.a.	n.a.	14.3 ± 6.7	14.9 ± 6.8

^ skewness <−1 or >1. n.a. = not applicable (not measured or not included in analysis).

**Table 2 jcm-09-04133-t002:** Moli-sani study dataset: prevalence of hyperphosphatemia, of high serum PTH, of low serum 1,25(OH)_2_D, of hypocalcemia, and of individuals with at least one of the CKD-MBD metabolic abnormalities in the group without reduced kidney function compared to the group with reduced kidney function (eGFR ≥ 90 and <90 mL/min × 1.73 m^2^, respectively).

	Men			Women		
	eGFR mL/min × 1.73 m^2^		Odds Ratio (95%CI)	eGFR mL/min × 1.73 m^2^		Odds Ratio (95%CI)
	≥90	<90		≥90	<90	
Number of examinees	195	293		139	328	
Hyperphosphatemia, %	2.1%	0.7%	0.33 (0.06/1.81)	5.8%	7.6%	1.35 (0.59/3.07)
High serum PTH, %	0.0%	2.0%	n.c.	0.7%	2.4%	3.45 (0.43/27.85)
Low serum 1,25(OH)_2_D, %	2.1%	3.8%	1.86 (0.58/5.94)	0.0%	3.7%	n.c.
Hypocalcemia, %	0.5%	1.4%	2.69 (0.30/24.2)	0.7%	0.6%	0.85 (0.08/9.41)
Anyone of the above, %	4.1%	7.2%	1.81 (0.78/4.16)	7.2%	13.4%	2.00 (0.98/4.10)

n.c. = not calculable.

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
