# Peer review of "Reduced Kidney Function and Relative Hypocalciuria—Observational, Cross-Sectional, Population-Based Data"

_jcm, 2020, doi:10.3390/jcm9124133_

Round 1

Reviewer 1 Report

In this manuscript the authors have reviewed the data of a large number of patients with varied eGFR and who had overnight and also fasting urine specimen to examine whether low urinary calcium excretion is correlated renal tubalar calcium reabsorption, PTH, and calcitriol and concluded that the most imported correlates with low urinary calcium/Cr is filtered calcium. I have the following suggestions:

  1. Given that the trust of the study was on the role of filtered calcium the investigators should have directly measured serum ultra filterable calcium concentration rather than calculated value.
  2. There is a mismatched in the number of patients with eGFR <90 ml/min.
  3. The lack of rigid dietary calcium intake also has been a signficant variable.
  4. I am not clear what is the difference between overnight fasting urine versus fasted voided urine. Do they consider overnight fast urine sample respresent post parandial calcium.
  5. There is no mention of the timming of last meal in these subjects which certainly affects overnight calcium levels.

Reviewer 2 Report

Jcm-1005360; Crillo et al

The Authors have investigated a possible relationship between urinary calcium excretion and impaired renal function by analysis of cross-sectional data from large on-going population studies in two defined regions of Italy following standard protocols. This adds to their series of publications from these studies. They found that urine calcium/creatinine ratios decreased with falling estimated glomerular function (eGFR). They suggest that the ratio might be a useful proxy of biochemical abnormalities in chronic kidney disease-mineral and bone disorder. The manuscript is clearly presented.

General Comments

1) My main concern is that analytical data for men and women have not been presented separately. It is well-established that women have lower plasma creatinine, urine creatinine excretion and creatinine clearance than men. Hence ratios for urine analytes related to creatinine are generally higher. The eGFR is lower even though factored for sex and could partially explain the higher proportion of women with low GFR observed, particularly in the Gubbio study in which eGFR was estimated from creatinine & not cystatin C as in the Moli-sani study.

The situation is more complicated, however, because women have lower calcium excretion than men, reflecting differences in bone turnover and intestinal absorption, and further, their calcium handling is different pre-& post-menopausally. This would reduce calcium/creatinine ratios. 

These physiologically based sex differences are evident in both studies (Supplementary Tables S2 & S3). The relative changes in calcium & creatinine may differ with renal impairment. This may be masked in the ANOVA comparisons for the full cohort, even though controlled for sex. Can data for men & women be shown separately in Table 1 in the text & Figs 1 &4 S2 & S3 & Table S6, and accompanying Anovas recalculated?

2) Despite the older age in the Moli-sani study serum creatinine concentrations were lower than for Gubbio, probably explained by use of a more accurate (enzymatic) assay for creatinine calibrated against an IDMS standard, not available in 1989-92 for the Gubbio study. This is relevant & should be mentioned since serum creatinine was used to estimate eGFR for Gubbio.

Specific comments

Introduction

P2 L 63; Prof Charles Pak (Texas) investigated the effects of calcium intake on urine calcium, comparing fasting & ingested loads

Methods

P2 L83 Was serum creatinine re-analysed? The IDMS standardisation with was not introduced until around 2006

P2 Para 1. Were stone formers excluded?- your earlier study demonstrated higher urine calcium excretion (Kidney Int 2003; 63: 2200) & this could confound the data analysis. Similarly, were other individuals with recorded disorders of calcium turnover excluded?

P3 L 138 In the cited reference (32), ultrafilterable calcium was estimated by first calculating the total calcium corrected for a low plasma albumin concentration (below 4g/dL): CaT corr = total Ca + 0.88 x [4 – Alb]. CaT corr was then multiplied by 0. 57 to obtain ultrafilterable calcium which is estimated at around 55-60% of total calcium. I am not clear how you calculated the value. Do you need to check your figures?

P3 L143 Typo: should be 8.6

P4 L158 There is a diurnal variation in creatinine clearance which is lowest during the night, hence the estimated 24h excretion is likely to be on the low side

Results

P4 &5. Table 1. What are the ranges shown? mean +/- ? SD; percentiles? For some of the indices which are clearly not normally distributed (eg first void urine  (Gubbio), medians & percentiles might be preferable. 

P6 L 200-204 Check that these linear regression findings still hold when men & women are considered separately.

P6 L207 Since eGFR was used to estimate the filtered calcium load it is not surprising that the two are linearly related

P7 L240 & Table 2 Serum phosphate is higher in healthy premenopausal women than men—how many women were in your ‘hyperphosphataemia’ group?

P8 L 267 Typo: should be 90 mL/min etc

Discussion

P9 L 286 Add ‘in’ after abnormalities

P9 L 287 Was there a statistically significant association between PTH & 1,25 (OH)2D?

P 9 Paras 2 & 3 You do not mention serum FGF23 which increases early in the course of renal impairment, reduces serum 1,25 (OH)2D and increases distal tubular calcium reabsorption (Wahl P, Wolf M. FGF23 in chronic kidney disease. Adv Exp Med Biol 2012;728:107-25; Musgrove J, Wolf M. Effects of FGF23 in chronic kidney disease. Ann Review Physiol 2020; 82: 365-90). It is likely explained by chronic hyperphosphataemia and raised PTH. This is an important omission which must be discussed

P9 L293- See above comment about higher serum phosphate in healthy pre-menopausal women. Do they account for the observed hyperphosphataemia?

P9 L 329-330 Considering the complex factors influencing urine calcium excretion in CRF, the Ca/crt would be a poor proxy for detecting low 1,25 (OH)2D but would be supportive evidence-as indicated in L 333.  Certainly it is indicated for monitoring treatment with calcium or 1,25 (OH)2D in patients with reduced renal function (L 335-337).

Supplementary

Tables S1, S2, S3: what are the ranges shown.

Table S3: check calculations of ultrafilterable calcium-see above

Tables S4 & S5: show data for men & women separately

Fig S1; To Legend add ‘estimated’ to same day 24h urine

Figs S2 & S3: show men & women separately (see above)

Round 2

Reviewer 1 Report

In the revised manuscript the authors responded to all of my inquiries. I feel at its present form the manuscript is suitable for publication.

Author Response

The authors did not provide a point-by-point response to reviewer 1 because he/she  wrote that "... the authors responded to all of my inquiries".

Reviewer 2 Report

M/S jcm-1005360-revised version 1

The Authors have addressed all my queries. Notably, they have re-analysed, & presented, all their data for men and women separately. This has demonstrated clear sex differences, & strengthened the paper substantially, without weakening the overall findings. As they have stated, population data relating urine calcium to renal function are sparse, & their data should now be useful to those running renal/ metabolic clinics.

In response to my query about whether individuals with stones or disturbed mineral metabolism were excluded, they have undertaken new multivariate analyses of the Gubbio data with these exclusions. There were small inconsistent differences from the original results but these are inconsequential. Reasonably, they have elected not to incorporate them into the revised manuscript.

FGF23 is now mentioned briefly in the Discussion, although I think it deserved a higher profile. Circulating FGF23 increases early in chronic renal failure & could well account for the reduction in 1,25 (OH)2 vitamin D and contribute to low urine calcium.  

Minor specific comments:

References

L63: A new ref 8 (Pak et al) was inserted-however:

L70: The original ref 8 has not been re-numbered, and the assignment of refs 9-11 seems to have gone awry-advise check (Methods para 1 L70-83)

Methods
L89. Should be slightly lower values for female (not male) sex prevalence

L126: more accurate to say ‘overnight Ca/creat ratio and the estimated ratio for 24 h urine’

Results

L241/242. True that 25(OH) vit D did not decrease with renal impairment, but Fig S4 shows a significant correlation (inverse); perhaps need to say this, then explain that this apparent increase lost significance after correcting for solar radiation?

Author Response

Reviewer’s comment

Response

The Authors have addressed all my queries. Notably, they have re-analysed, & presented, all their data for men and women separately. This has demonstrated clear sex differences, & strengthened the paper substantially, without weakening the overall findings. As they have stated, population data relating urine calcium to renal function are sparse, & their data should now be useful to those running renal/ metabolic clinics.

The authors fully agree. The reviewer’s suggestion of separate analyses for men and women improved substantially the presentation of results.

In response to my query about whether individuals with stones or disturbed mineral metabolism were excluded, they have undertaken new multivariate analyses of the Gubbio data with these exclusions. There were small inconsistent differences from the original results but these are inconsequential. Reasonably, they have elected not to incorporate them into the revised manuscript.

The authors agree.

FGF23 is now mentioned briefly in the Discussion, although I think it deserved a higher profile. Circulating FGF23 increases early in chronic renal failure & could well account for the reduction in 1,25 (OH)2 vitamin D and contribute to low urine calcium.

The authors do agree that FGF23 has a role in changes of urine calcium and 1,25(OH)2D due to chronic kidney failure. However, they decided not to expand the text about FGF23 because the paper did not include any data about this variable.

L63

L70

A new ref 8 (Pak et al) was inserted-however:

The original ref 8 has not been re-numbered, and the assignment of refs 9-11 seems to have gone awry-advise check (Methods para 1 L70-83)

The L70 error in reference number was corrected.

L89

Should be slightly lower values for female (not male) sex prevalence

The typo was corrected in the second revision.

L126

more accurate to say ‘overnight Ca/creat ratio and the estimated ratio for 24 h urine’

The text was modified as per reviewer’s suggestion.

L241

/

L242

True that 25(OH) vit D did not decrease with renal impairment, but Fig S4 shows a significant correlation (inverse); perhaps need to say this, then explain that this apparent increase lost significance after correcting for solar radiation?

The authors agree but preferred not to expand the text about this point because 25(OH)D was not the main objective of the paper.  Actually, the list of 25(OH)D correlates will likely be the focus of a future paper including not only solar radiation but also dietary vitamin D intake, serum albumin, serum lipids, adiposity, and others.
